# Scalable Influence and Fact Tracing for Large Language Model Pretraining

**Tyler A. Chang,**[1,2][*] **Dheeraj Rajagopal,**[1] **Tolga Bolukbasi,**[1] **Lucas Dixon,**[1] **Ian Tenney**[1]

`tachang@ucsd.edu, {rajagopald,tolgab,ldixon,iftenney}@google.com`

[1]Google DeepMind    [2]UC San Diego

## Abstract

Training data attribution (TDA) methods aim to attribute model outputs back to specific training examples, and the application of these methods to large language model (LLM) outputs could significantly advance model transparency and data curation. However, it has been challenging to date to apply these methods to the full scale of LLM pretraining. In this paper, we refine existing gradient-based methods to work effectively at scale, allowing us to retrieve influential examples for an 8B-parameter language model from a pretraining corpus of over 160B tokens with no need for subsampling or pre-filtering. Our method combines several techniques, including optimizer state correction, a task-specific Hessian approximation, and normalized encodings, which we find to be critical for performance at scale. In quantitative evaluations on a fact tracing task, our method performs best at identifying examples that *influence* model predictions, but classical, model-agnostic retrieval methods such as BM25 still perform better at finding passages which explicitly contain relevant facts. These results demonstrate a misalignment between factual *attribution* and causal *influence*. With increasing model size and training tokens, we find that influence more closely aligns with factual attribution. Finally, we examine different types of examples identified as influential by our method, finding that while many directly entail a particular fact, others support the same output by reinforcing priors on relation types, common entities, and names. We release our prompt set and model outputs, along with a web-based visualization tool to explore influential examples for factual predictions, commonsense reasoning, arithmetic, and open-ended generation for an 8B-parameter LLM.[1]

## 1 Introduction

Modern large language models (LLMs) perform extraordinarily well at a wide variety of natural language tasks, but exactly how they leverage training data to achieve such capabilities is not well understood. One promising avenue of research to study this is *training data attribution* (TDA), which aims to identify influential training examples for given model predictions. When successful, TDA can serve as a method to both inspect and intervene on the training process. For example, it can enable training data curation targeted at specific tasks (Engstrom et al., 2024), reduction of data contamination (Mozes et al., 2023), and better transparency into model predictions (Grosse et al., 2023; Choe et al., 2024).

However, the steep computational cost of applying TDA approaches to LLMs has limited work in this area. TDA approaches have achieved promising results identifying examples that influence LLM behavior during fine-tuning (Akyurek et al., 2022; Park et al., 2023; Xia et al., 2024), but there is significant evidence that much of an LLM's knowledge and capabilities originates from pretraining (Hoffmann et al., 2022; Chang & Bergen, 2024). Previous work applying TDA to pretraining has either focused on small models (Engstrom et al., 2024) or extremely few target queries, e.g. retrieving influential examples from a significantly subsampled corpus for less than 100 model predictions (Grosse et al., 2023; Choe et al., 2024; Ruis et al., 2024). In our work, we scale TDA experiments to LLMs up to 8B parameters, thousands of model predictions, and corpora up to 160B tokens.

---

[*]Work done as a student researcher at Google Research and Google DeepMind.
[1]`https://github.com/pair-code/pretraining-tda`

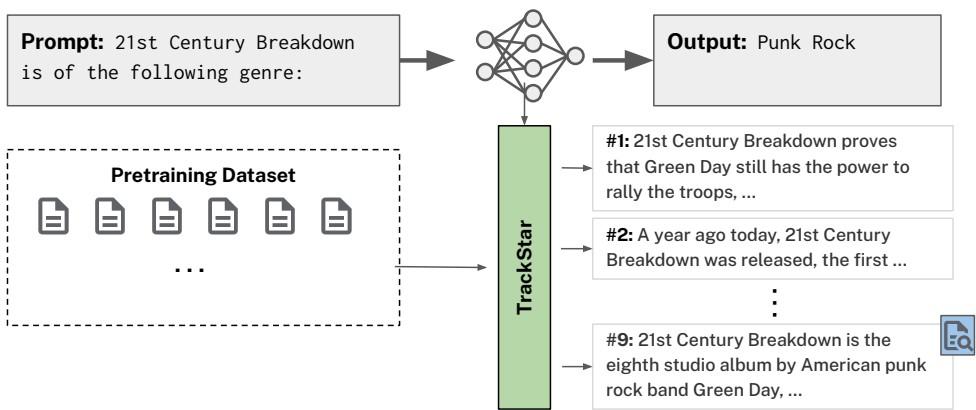

Figure 1: Top proponents from C4 using TrackStar given a factual query and model prediction. TrackStar is a gradient-based method that approximates *influence* on the model, which we show may not always be optimal for *attribution*, which involves finding examples which directly entail the target factual prediction.

Specifically we propose TrackStar, a gradient-based influence method that combines innovations from previous work and scales to large setups (§3), while still supporting efficient retrieval of influential examples and quantitative evaluation. At scale, our method significantly outperforms previous influence methods at retrieving pretraining examples that entail a fact (*attribution*) and examples that influence specific factual predictions (*influence*) (§5, §6). Importantly, we demonstrate a misalignment between attribution and influence; classical methods such as BM25 are better at retrieving examples that entail factual predictions, but those examples are not necessarily those that most affect model predictions (§5.1). We show that influence grows closer to attribution as models scale, both in parameters and training tokens (§5.3).

To provide insights into where attribution and influence misalign, we include analyses of examples that have high influence for factual predictions in LLMs (§7). For example, rather than containing a full statement of a fact of interest, many examples appear to support priors on relation types, common entities, or names.

## 2 BACKGROUND: TRAINING DATA ATTRIBUTION

Researchers have proposed a variety of methods to measure the influence of individual training examples on output model metrics (e.g. loss on target datasets). Some of the best results come from simulation-based methods such as Datamodels (Ilyas et al., 2022) and Simfluence (Guu et al., 2023), which estimate contributions based on multiple training runs with different data subsets. However, this is not tractable for LLM pretraining, which is expensive to perform even once. For computational tractability, we focus on *gradient-based methods* that use parametric approximations, based on a single model, to predict how a model's behavior would change under the removal (or addition) of specific training examples.

To quantify the influence from a training example $z_m$ to an evaluation (query) example $z_q$, most gradient-based influence methods compute some version of a Hessian-corrected dot product $-\nabla L(z_q) H^{-1} \nabla L(z_m)$ of model gradients for $z_m$ and $z_q$ loss (Koh & Liang, 2017; Schioppa et al., 2022; Park et al., 2023; Grosse et al., 2023). Here, $H^{-1}$ is the inverse Hessian matrix of the loss with respect to model parameters. The resulting dot product approximates changes in $z_q$ loss from adding example $z_m$ during training. In practice, existing methods use a variety of methods to approximate the Hessian $H$, notably the autocorrelation matrix $\tilde{\Phi}^T \tilde{\Phi} \in \mathbb{R}^{|\theta| \times |\theta|}$ of per-example gradients in TRAK (Park et al., 2023; Sagun et al., 2018) or the closely-related approximate curvature matrix $G$ of EK-FAC (Grosse et al., 2023). These methods also tie closely back to TracIn (Pruthi et al., 2020), which computes gradient dot products aggregated over model checkpoints; recent work using TracIn has included gradient second moment correction (Akyurek et al., 2022; Xia et al., 2024), equivalent to a diagonal Hessian approximation (§3).

However, when applying influence methods to LLM pretraining, an additional bottleneck is the scale of model parameters and pretraining data. Given $|\theta|$ model parameters (on the order of billions), we have loss gradients $\nabla L(z) \in \mathbb{R}^{|\theta|}$ and inverse Hessian $H^{-1} \in \mathbb{R}^{|\theta| \times |\theta|}$. To rank pretraining examples, the gradient dot product must be computed between every pretraining example $z_m$ (on the order of millions to billions) and the target query $z_q$. For this reason, previous work has focused on identifying influential examples during fine-tuning (Akyurek et al., 2022; Park et al., 2023; Xia et al., 2024) or during pretraining for smaller models (Engstrom et al., 2024). Closest to our work has been that of Grosse et al. (2023), Choe et al. (2024), and Ruis et al. (2024), who look at LLMs with billions of parameters; however, they perform retrieval on a small subset of the pretraining corpus and report evaluation on only a small number of queries.

## 3 METHOD: TRACKSTAR

Here, we introduce TrackStar, a gradient-based influence method that combines innovations from previous work that scales effectively to large settings. Following the gradient-based methods in §2, we compute the influence between two examples as the dot product between the projected and corrected model gradients for those examples. Because correction terms vary substantially in previous work, here we describe the motivation behind our specific approach, with ablation experiments in §5.2. Specifically, given a query $z_q$ (an input prompt with corresponding model completion or desired completion) and a training example $z_m$, we compute the influence from $z_m$ to $z_q$ as:

$$\mathcal{I}_\theta(z_m, z_q) = \bar{G}_\theta(z_m) \cdot \bar{G}_\theta(z_q) \tag{1}$$

where $\bar{G}_\theta(z)$ is the projected, Hessian-corrected, and unit normalized gradient for example $z$ given model parameters $\theta$:

$$\bar{G}_\theta(z) = \frac{G_\theta(z)}{||G_\theta(z)||_2} \qquad G_\theta(z) = R^{-\frac{1}{2}} P_d \frac{\nabla_\theta \text{Loss}(z, \theta)}{\sqrt{V}} \tag{2}$$

In turn, we define:

- **Loss gradient** : $\nabla_\theta \text{Loss}(z, \theta) \in \mathbb{R}^{|\theta|}$: As in previous work, we compute the loss gradient for each example $z$ with respect to model parameters $\theta$ (Pruthi et al., 2020; Akyurek et al., 2022; Han & Tsvetkov, 2022; Grosse et al., 2023; Xia et al., 2024). The original TRAK method uses the multi-class margin function gradient; we evaluate different output functions in §A.2.1, but we find that loss gradients perform best empirically. For each example $z$, we sum gradients over target tokens; for training examples, this comprises all tokens in the example. If an input prompt is given, we only include tokens in the model completion. In contrast to previous work that selects only a subset of layers (Akyurek et al., 2022; Yeh et al., 2022; Grosse et al., 2023), we compute gradients with respect to all model parameters $\theta$ except the token embedding layer, pooled into layer blocks (§A.2.2) before dimensionality reduction with random projection.

- **Second moment estimate** : $V \in \mathbb{R}^{|\theta|}$: To account for high-magnitude gradient components that might dominate gradient dot products (e.g. for outlier model components; Timkey & van Schijndel, 2021; Puccetti et al., 2022), we correct by an estimate of the expected magnitude of the loss gradient with respect to each model parameter. Formally, for each parameter $x \in \theta$, this is an estimate of the second moment $\mathbb{E}_z((\nabla_x \text{Loss}(z, \theta))^2)$ of the gradient with respect to $x$. These estimates are used by common optimizers such as Adafactor (Shazeer & Stern, 2018) and Adam (Kingma & Ba, 2015), and as such the gradients corrected by $V$ can be seen also as more faithful to the model's training process (Pruthi et al., 2020). We use the estimates of $V$ computed by Adafactor, which can be efficiently applied by element-wise multiplication.

  Notably, dividing by the square root second moment $\sqrt{V}$ is equivalent to using only the diagonal of the Gauss-Newton Hessian approximation (gradient autocorrelation matrix $R$ around zero; Sagun et al., 2018) from previous work (§2). Unlike TRAK (Park et al., 2023; Engstrom et al., 2024), which corrects gradients by the autocorrelation after random projection, the optimizer second moment estimates allow us to apply the correction per parameter before random projection, enabling more granular correction of individual outlier components.

- **Random projection** : $P_d \in \mathbb{R}^{d \times |\theta|}$: As in TRAK (Park et al., 2023; Engstrom et al., 2024) and as described in the original TracIn paper (Pruthi et al., 2020), we use Gaussian random projection to reduce the dimensionality of the full model gradients. To improve projection efficiency, we use

the two-sided projection from Pruthi et al. (2020), which is equivalent to the recently proposed LoGra method of dimensionality reduction (Choe et al., 2024; §A.2.2). This approach contrasts with Grosse et al. (2023), who use either un-projected model gradients or query-specific low-rank reductions. Un-projected model gradients are too large to store for all pretraining examples, and query-specific approximations require that all pretraining example gradients be re-computed if a new query is considered. Random projections allow the projected gradient to be computed and saved exactly once per pretraining example, usable for all future retrievals. We use projection dimensionality $d = 2^{16}$, but we experiment with lower dimensionalities in §5.2.

- **Hessian approximation** : $R \in \mathbb{R}^{d \times d}$: We follow Park et al. (2023) and Engstrom et al. (2024) in using the autocorrelation matrix $R = \tilde{\Phi}^T \tilde{\Phi}$ of per-example gradients as a Gauss-Newton approximation to the loss Hessian. We compute and apply this after optimizer state correction and random projection. For efficiency, we enforce a block-diagonal structure (§A.2.3) which allows us to efficiently compute $R^{-\frac{1}{2}}$ (Eq. 2). However, using this method, we still find that retrievals suffer from common proponents that largely mimic the task template. Thus, departing from previous work, we consider a mixing approach where the matrix $R_{\text{train}}$ estimated from pretraining example gradients is mixed with $R_{\text{eval}}$ derived from evaluation example gradients:

$$R = \lambda R_{\text{eval}} + (1 - \lambda) R_{\text{train}} \qquad (3)$$

This mixture allows $R^{-\frac{1}{2}}$ to downweight high-magnitude components that are common for evaluation examples in a task, such as components corresponding to the task template. We select the mixing parameter $\lambda$ such that the top ~1000 task-specific gradient components (out of 65K) are downweighted (details in §A.2.3). For T-REx closed set experiments (§5), we use $\lambda = 0.90$; for C4 open set experiments (§6), we use $\lambda = 0.99$.[2]

- **Unit normalization**: To compute cosine similarity, we unit normalize both input vectors in Equation 1, as in Akyurek et al. (2022), Han & Tsvetkov (2022), Choe et al. (2024), and Xia et al. (2024). This reduces the effect of outlier training examples that have high overall gradient magnitudes (Barshan et al., 2020; Han & Tsvetkov, 2021) and thus appear as common proponents before unit normalization. Unit norming is equivalent to $\ell$-relative influence (Barshan et al., 2020), which identifies training examples that maximize the query example loss change while constraining the overall loss change.

Using the dot product of unit normed vectors $\bar{G}_\theta(z)$ (Eq. 2) to quantify the influence $\mathcal{I}_\theta(z_m, z_q)$ of each training example $z_m$ on query $z_q$, we are able to retrieve the top $k$ training examples for $z_q$. We refer to these top-ranking examples as proponents of $z_q$ (Pruthi et al., 2020). In the spirit of TracIn and TRAK, we refer to our proponent ranking method as Trac*, or "TrackStar".

## 4 MODELS, DATASETS, AND METRICS

We apply TrackStar to identify proponent examples for factual predictions in LLMs up to 8B parameters. Unless otherwise specified, we first build an index of the projected gradients for all candidate examples of interest (e.g. the pretraining set), then compute the Hessian approximation $R$ using a single pass over the data. For gradient-based methods, we perform exact scoring between each query and all candidate examples, with no need for lexical pre-filtering (c.f. Akyurek et al., 2022; Park et al., 2023; Grosse et al., 2023). All elements of this approach have compute cost linear in model and dataset size (§2).

### 4.1 MODELS

We pretrain a decoder-only language model on two epochs of English C4 (Raffel et al., 2020) for three model sizes: 154M, 1B, and 8B parameters, using the architecture described in Chowdhery et al. (2023). Using our tokenizer, English C4 consists of 160B tokens across 365M examples (short passages and documents). Model details are in §A.3. We focus primarily on the 8B model, but we study the smaller models for dimensionality ablations (§5.2) and scaling (§5.3).

---

[2]Described in §A.2.3, C4 requires larger $\lambda$ because the pretraining example gradients tend to be larger than those for T-REx sentences, due to longer sequence lengths.

## 4.2 Fact Tracing Dataset

We focus on identifying proponent examples for factual predictions in LLMs. For factual recall prompts, we use the filtered T-REx dataset from KILT (Petroni et al., 2021), which consists of entity-relation-entity triples such as (*Carleton College*, *country*, *USA*). We merge the KILT dataset back into the original T-REx dataset (Elsahar et al., 2018), obtaining a dataset of facts with corresponding entity IDs, surface form aliases, and Wikipedia abstract sentences containing each fact.[3] For each fact, there are an average of 2.5 "ground-truth" entailing sentences out of 19.6M sentences total. After filtering out ambiguous prompts (e.g. multiple correct target entity IDs), we have a total of 1.2M fact triples covering 97 relation types. Following Kandpal et al. (2023), we also annotate facts with the count (fact frequency) of pretraining examples from C4 that mention both entities.

We manually write natural language templates for all 97 relation types, so that a left-to-right LLM can predict the target fact as a completion, e.g. "*Carleton College is located in the following country: USA*". We mark predictions as correct using string matching after lowercasing and stopword removal, considering multiple possible correct surface form aliases as provided by KILT. Random chance accuracy on this task is close to 0% due to the large number of possible completions; our 154M, 1B, and 8B models achieve 16.3%, 28.0%, and 32.4%, respectively. For factual queries in all following evaluations, we use a fixed sample of 5415 facts drawn randomly from this set but balanced for fact frequency and down-sampling common target entities (such as *"USA"* or *"English"*), and limiting facts that are incorrectly predicted by all models. Each query contains an input prompt and ground truth target text. Dataset details and pre-processing are described in §A.4.

## 4.3 Evaluation Metrics

After retrieving proponents for a given query, we consider traditional fact tracing metrics (MRR and recall, *attribution*; Akyurek et al., 2022) and a tail-patch metric that quantifies the effect of proponents on the model itself (*influence*).

**Attribution metrics: MRR and Recall@10.** These measure whether we retrieve examples that logically support (entail) a fact. For T-REx evaluations (§5), entailing Wikipedia abstract sentences are annotated in the dataset (§4.2). For C4 evaluations (§6), because passages are not annotated, we use an entailment model to score whether a candidate passage contains factual information supporting the query. Specifically, we use a fine-tuned T5 11B (Raffel et al., 2020) model trained on ANLI (Nie et al., 2020) and synthetic data as described in Gekhman et al. (2023). Because C4 passages can be up to 2048 tokens, we split the input C4 passages by sentences using regex matching, and we take the maximum entailment score using a sliding window of three sentences as the input premise and the factual query as the hypothesis. We mark a proponent as entailing when this entailment score $\geq 0.50$.[4] For MRR, we compute the mean reciprocal rank of the top-ranked "entailing" proponent for each fact. For recall@10, we compute the proportion of facts for which an entailing proponent appears in the top 10 proponent retrievals.

**Influence metric: incremental training probability increase (tail-patch).** Both of the previous metrics assume that the top-ranked proponents should be those that *entail* the target fact. However, these proponents are not necessarily the examples that would most influence the model to make that prediction. Following Koh & Liang (2017), most work on influence methods derives metrics from leave-one-out or other ablate-and-retrain experiments such as linear datamodeling score (Ilyas et al., 2022). However, this is not tractable for full-scale LLM pretraining, so we instead estimate the *additive* contribution from incremental training. Starting from the final model checkpoint and maintaining all pretraining hyperparameters, we take a single training step—which we call a *tail-patch* step—on a single retrieved proponent and measure the change in probability of the target sequence. We then average this change across the top $k = 10$ proponents for each query, and across all queries in the evaluation set. Results for different $k$ are reported in §A.5.2.

---

[3]Because T-REx is automatically scraped, there is some inherent noise in the "ground truth" labels.

[4]Based on blinded manual annotation of 100 C4 passages marked as entailing and non-entailing respectively, we find a false positive rate of 13% and a false negative rate of 11%. Of the incorrect annotations, most (54%) are fuzzy entailments where a passage implies but technically may not entail a fact.

## 4.4 BASELINE METHODS

We compare four ranking methods to retrieve proponent examples: **BM25** (Kamphuis et al., 2020), **Gecko** embeddings (Lee et al., 2024), **TRAK** (Park et al., 2023; Engstrom et al., 2024), and our method, **TrackStar**. As described in §5.2, we also implicitly compare to methods from several previous studies by ablating different correction terms from TrackStar. As a classical retrieval baseline, BM25 is a bag-of-words method that ranks candidates according to query terms appearing in each document, with TF-IDF weighting ("Lucene accurate" version; Kamphuis et al., 2020). Gecko is a text embedding model trained with a two-step distillation procedure and contrastive loss (Lee et al., 2024).[5] TRAK is a gradient-based influence method that has been shown to perform well in previous work; the recommended version of TRAK in Park et al. (2023) and Engstrom et al. (2024) is similar to TrackStar but uses the multi-class margin gradient and a non-task-specific Hessian approximation, with no optimizer second moment correction or unit norm.[6]

## 5 T-REX CLOSED SET EVALUATION

We first evaluate TrackStar at fact tracing and influence on the T-REx dataset (§5.1), and we find that it outperforms prior gradient-based TDA methods (§5.2). For each query we retrieve the top 10 highest scoring candidate sentences from the set of 19.6M Wikipedia abstract sentences in T-REx.

### 5.1 T-REX CLOSED SET RESULTS

In Table 1, our results show that TrackStar outperforms other variants at identifying proponent examples for both attribution (identifying fact-entailing proponents) and influence (tail-patching scores) in the 8B model. Notably, previous work (Akyurek et al., 2022; Park et al., 2023) on this task considers only small candidate sets of 300 "distractors" per query, but we retrieve from the entire set of 19.6M sentences. We observe that this makes a significant difference in performance: while our TRAK replication achieves an MRR of $0.401$ when replicating the smaller setting (similar to as reported by Park et al., 2023), it only achieves an MRR of $0.001$ (and *negative* tail patch scores) in the full setting evaluated here. We find that this poor performance is largely due to the use of multi-class margin function gradients with correction applied per example rather than per token (§A.2.1), along with the lack of unit normalization (Table 1, Experiment 1 vs. Experiment 2). This result highlights that methods that perform well in small settings and classification tasks may be susceptible to noise in larger settings with LLMs.

We also find that classical, model-agnostic retrieval approaches (BM25 and Gecko) are significantly better than gradient-based influence methods at attribution (i.e. retrieving proponents that entail a given fact; MRR and recall; Table 1). Despite this, proponents from these classical methods still have *much less influence* on model predictions than proponents from influence methods. Tail-patching on proponents from BM25 (tail-patch score $+0.41\%$) increase target fact probabilities by $2.2\times$ less than proponents from TrackStar (tail-patch score $+0.90\%$). Even tail-patching on "ground truth" entailing sentences from T-REx (tail-patch score $+0.52\%$) results in much smaller probability changes than proponents from TrackStar.[7] This result suggests a distinction between *attribution* and *influence*; while classical, model-agnostic retrieval methods perform well at attribution (which we note is highly lexical and well-suited for string matching methods such as BM25; Wang et al., 2021), influence methods better predict how a proponent might affect the model and may better reflect the model's reasoning (§7).

### 5.2 ABLATION EXPERIMENTS

**Correction terms**: In Table 1, we conduct ablations (labeled by experiment numbers) to determine the source of improvements for TrackStar over other gradient-based influence methods. Many of these ablations are close or equivalent to methods from previous work. For example, Experiment 1

---

[5]We use the $d = 768$ embedding model `textembedding-gecko@003` available on Google Cloud.

[6]TRAK has one additional difference, in that it multiplies scores by $Q = 1 - \bar{p}$, where $\bar{p}$ is the probability of the candidate sequence, averaged over tokens. We find $Q$ ranges from 0.6 to 1.0 across candidate sequences, and it has little effect on MRR, recall, and tail-patch scores. Theoretical motivations are described in §A.2.1.

[7]This echoes results from Park et al. (2023) for fact fine-tuning.

| Exp. # | Optim. | $R$ | Unit norm | MRR | Recall@10 | Tail-patch |
|--------|--------|-----|-----------|-----|-----------|------------|
| T-REx gold | – | – | – | Gold | Gold | **+0.52%** |
| BM25 | – | – | – | 0.592 | 0.773 | +0.41% |
| Gecko | – | – | ✓ | **0.620** | **0.794** | +0.31% |
| TRAK | | ✓* | | 0.001 | 0.001 | –0.02% |
| 1 | | | | 0.064 | 0.114 | +0.35% |
| 2 | | | ✓ | 0.266 | 0.358 | +0.65% |
| 3 | | ✓ | ✓ | 0.290 | 0.399 | +0.85% |
| 4 | ✓ | | ✓ | 0.300 | 0.413 | +0.71% |
| 5 | ✓ | ✓ | ✓ | 0.295 | 0.406 | +0.87% |
| TrackStar | ✓ | Mixed | ✓ | **0.365** | **0.496** | **+0.90%** |

Table 1: Results on T-REx closed set evaluation for the 8B-model (§5). Our setup is significantly more difficult than prior work, retrieving from all 20M T-REx abstract sentences rather than 300 "distractors" per fact. Classical retrieval results are reported in the top section for comparison. We note TRAK uses multi-class margin function gradients rather than loss gradients; details in §A.2.1.

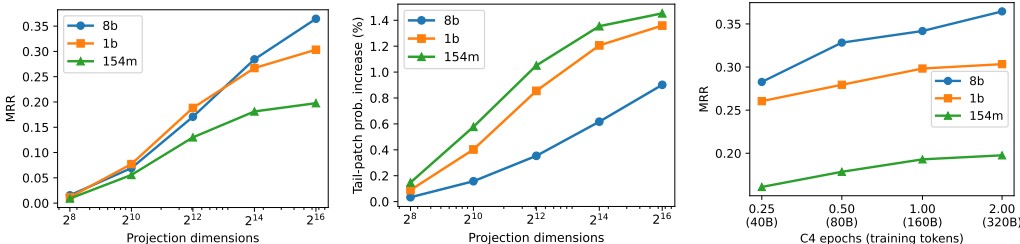

Figure 2: Left, center: attribution (MRR) and influence (tail-patch) scores as a function of gradient projection dimensionality $d$ for different model sizes (§5.2). Right: attribution (MRR) scores throughout pretraining for different model sizes. As models improve, TrackStar influence becomes more similar to attribution (higher MRR; §5.3).

corresponds closely to Pruthi et al. (2020), without the use of multiple checkpoints, while Experiment 2 corresponds to Han & Tsvetkov (2022) and Han et al. (2023). Experiment 3 is the method of Choe et al. (2024), and Experiment 4 corresponds to Akyurek et al. (2022) and Xia et al. (2024).[8]

First, we find that unit normalization is key to good MRR, recall, and tail-patch scores (Experiments 1–2; also verified by ablating unit normalization from other experiments). Optimizer second moment correction provides additional improvement, similar to including the Hessian approximation $R$ (Experiments 3–4); intuitively, both methods perform Hessian correction by approximating the Hessian with gradient variance terms (§3). Using optimizer correction and $R$ together produces fairly similar results to either optimizer correction or $R$ alone (Experiments 3–5). In any case, the mixed task-specific Hessian approximation $R$ (downweighting common gradient components for the fact completion task; §3) provides substantial improvements, particularly for MRR and recall.

In fact, on a theoretical basis, it may be somewhat surprising that unit normalization and task-specific Hessian approximation improve tail-patch scores at all. These two normalizations encourage the scoring method to focus on proponents specific to the target fact by downweighting proponents that affect loss overall (or loss for the overall task). These downweighted proponents might still be expected to have high tail-patch scores (large effects on target fact probability), despite their significant effect on loss overall, because tail-patch scores do not constrain the overall loss change induced by a proponent. The fact that these downweighted proponents actually have lower tail-patch scores (i.e. lower tail-patch scores for proponents before unit normalization and task-specific Hessian correction) indicates that these corrections actually have an overall denoising effect. Despite their motivation based on maximizing target influence under overall loss change constraints, these corrections actually find proponents that maximize target influence even without such constraints.

**Projection dimensionality**: Higher gradient projection dimensionality $d$ results in higher fidelity representations, but both the memory required to store projected gradients and the retrieval cost to

---

[8]Xia et al. (2024) also normalize by Adam first moment estimates, but these are not available in Adafactor.

| Method | MRR | Recall@10 | Tail-patch |
|---|---|---|---|
| BM25 | **0.687** | **0.845** | +0.83% |
| Gecko | 0.636 | 0.826 | +0.54% |
| GRADIENT DOT PRODUCT | 0.003 | 0.015 | +0.04% |
| GRADIENT COSINE | 0.252 | 0.393 | +1.95% |
| TrackStar | 0.338 | 0.515 | **+2.11%** |

Table 2: Results retrieving proponents from all of C4 (§6). GRADIENT COSINE ablates the Hessian approximation $R$ from TrackStar, and GRADIENT DOT PRODUCT further ablates unit normalization.

compute dot products increase (linearly) with $d$. To balance these considerations, in Figure 2 (left, center), we plot how attribution (MRR) and influence (tail-patch) scores improve with projection dimensionality for TrackStar.[9] For smaller models (154M and 1B parameters), scores have begun to plateau around $d = 2^{16}$. Although the 8B-parameter model has not yet plateaued, we restrict our experiments to $d = 2^{16}$ due to memory limitations, as the index of 365M C4 examples at $d = 2^{16}$ is already 87TB in size (§6). Furthermore, we note that our factual knowledge task likely requires a large number of dimensions due to the inherent difficulty of compressing individual facts about a large number of entities; fewer dimensions may be sufficient for other tasks.

### 5.3 INFLUENCE APPROACHES ATTRIBUTION

In §5.1, we demonstrated a distinction between *attribution* (identifying proponents that entail a fact, quantified by MRR and recall) and *influence* (identifying proponents that influence the model prediction, quantified by tail-patch scores). We find that TrackStar attribution scores improve as models increase in both parameters and training tokens (Figure 2, right). This indicates that as models improve, the influential examples that TrackStar identifies align more with attribution, suggesting that more capable models rely more on examples that actually entail facts for factual predictions. Of course, it does not appear that these measures converge entirely; even for our largest model, tail-patching on ground truth entailing proponents results in much smaller target probability changes than TrackStar proponents (§5.1). Thus it appears that as models improve, they are more likely to use entailing examples to learn facts, but there are still many proponents that have large effects on the model despite non-entailment. We present a deeper analysis of these types of proponents in §7.

## 6 C4 OPEN SET EVALUATION

While the T-REx setting is useful for controlled experiments, in practical use LLMs are trained from large, unsupervised corpora containing a wide variety of passages that the model may learn from. To extend TDA to this setting, we apply TrackStar to identify influential examples for our 8B-parameter model from all 365M passages in the C4 corpus. In this scenario, our candidate set (C4) consists of all training examples that the LLM has ever seen. These candidate sequences are often much longer than T-REx sentences (up to 2048 tokens), and they cover many domains rather than just Wikipedia.

For the same set of 5.4K factual queries with ground truth targets as §5, we retrieve proponents using TrackStar, BM25, and Gecko. As C4 is a much larger corpus and is proportionally more expensive to compute gradients over, we perform a more limited set of ablations as shown in Table 2. We focus on the two corrections that had the largest effects in §5.2: mixed task-specific Hessian approximation $R$ (ablated in GRADIENT COSINE in Table 2) and unit normalization (further ablated in GRADIENT DOT PRODUCT in Table 2).[10] Both of these ablated methods still significantly outperform original TRAK in §5. As before, we quantify performance using MRR, recall, and tail-patch scores.

### 6.1 C4 OPEN SET RESULTS

In line with §5.1, TrackStar has better tail-patch (*influence*) scores than all other methods, and better MRR and recall (*attribution*) than other gradient methods (Table 2). Again, we find that unit

---

[9]Park et al. (2023) find that performance may decrease if projection dimensionality is too large, but our experiments do not appear to be close to that limit, likely due to the high dimensionality $|\theta|$ of LLM gradients.

[10]GRADIENT COSINE is equivalent to Experiment 4 in Table 1, and GRADIENT DOT PRODUCT is equivalent to Experiment 1 with optimizer second moment correction.

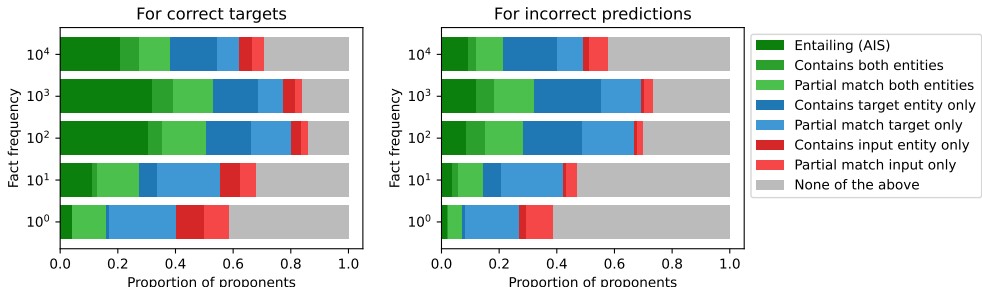

Figure 3: Proportions of TrackStar proponents (top 10 per query) retrieved from C4 that entail a prediction, contain both entities, or contain only one entity for a model prediction.

normalization (cosine) is critical to performance, and the addition of task-specific Hessian approximation $R$ in TrackStar further improves results. In particular, lack of unit normalization often leads to retrieval of long irrelevant examples, as these tend to have large gradient magnitudes.

Also in line with §5, BM25 and Gecko perform much better than TrackStar for attribution (MRR 0.687 and 0.636 vs. 0.338). Still, we note overall high MRR and recall scores given the difficulty of the task: for 51.5% of facts, TrackStar retrieves an entailing example in the top 10 out of 365M C4 examples. Additionally, TrackStar performs over $2.5\times$ better than BM25 and Gecko at retrieving examples that influence model predictions (tail-patch score $+2.11\%$ vs. $+0.83\%$ and $+0.54\%$). This reinforces the distinction between attribution and influence: examples that entail facts—such as proponents retrieved by BM25—are often not the examples that most affect a model's predictions.

## 7 HEADROOM ANALYSIS

In §5 and §6 we showed that TrackStar makes significant progress towards attribution for LLM pretraining, outperforming other methods at retrieving *influential* pretraining examples, but it still falls short of classical baselines on the fact tracing task (*attribution*). To better understand why, we look in more detail at the types of proponents retrieved by our method. We find that much of the headroom can be explained by proponents which reflect priors or partial matches, multi-hop reasoning, or idiosyncrasies of the fact tracing task such as ambiguous entities or alternative correct answers. Examples of these proponents are included in Table A.1 and Table A.2. The full set of proponents can be viewed at `https://github.com/pair-code/pretraining-tda`, including proponents for additional evaluation tasks as described in §A.6.

**Priors and partial matches**: In an ideal world, model predictions would be informed by examples that fully describe (entail) that fact (Chang et al., 2024), and these would appear as the top proponents. However, the model's output can also be informed by priors, such as the probability that the answer to any question about language being "*English*" or a city being "*New York*". In Figure 3 (left), we examine the distribution of proponents (top 10 per query) from TrackStar based on their relation to the query: (1) a full entailment (as in §6), (2) containing the input and target entity but without entailing the fact, (3) containing only one of the entities, or (4) containing neither entity. The latter three categories are estimated by string-matching, and we also consider partial-matches where the proponent matches at least one non-stopword of an entity (e.g. a last name). Additionally, we stratify these categories by how frequently the fact appears in the pretraining corpus.

We find that a majority of proponents fall into one of the full- or partial-match categories, with close to 40% of proponents entailing the query on moderate-frequency ($10^2$ to $10^3$ occurrences in C4) facts and 40% more with at least a partial match to one of the entities. While it is not clear why partial matches sometimes score more highly than entailing examples, it is plausible that especially when the target entity is mentioned, these examples would contribute to the model's prior $P(y)$ of generating the target string; 62% of proponents contain at least a partial match to the target entity. For less common entities (e.g. less frequent facts), partial matches appear more frequently, such as matching the first name in *"Alemayehu Shumye"* in Table A.1. These may represent fallback reasoning, where the model does not know about the specific entity but makes a guess based on statistical associations, such as between names and nationalities.

Interestingly, we observe a drop in entailing and entity-matching proponents for the highest frequency facts ($\geq 10^4$ occurrences). In part, this appears due to data artifacts, as some examples in T-REx are string matches for common nouns, such as *"City is located in the following country: United States"*, and it is unclear how a model *should* interpret these queries. Additionally, there may be saturation effects (Pruthi et al., 2020), where these common facts (such as that Munich is a city in Germany in the *"Munich Symphony Orchestra"* example in Table A.1) are learned early in training, and gradients at the final checkpoint are diminished.

**Multi-hop reasoning**: We also observe some cases where there is a multi-step reasoning path between a proponent text and the query. For example in the *"Carel Steven Adama van Sheltema"* example in Table A.1, a prompt asks about a person's native language, and the first proponent passage states that they were born in Amsterdam. While not strictly entailing, this does provide support for the model to plausibly guess that the language is *"Dutch"*.

**Noisy proponents**: However, some retrieved proponents are simply noisy. In a relatively small number of cases, examples are entirely irrelevant (e.g. *"Cadillac"* in Table A.1). In these cases, we often find that TrackStar and other gradient-based methods retrieve long or repetitive passages which bear minimal relation to the query, or which repeat the target string many times (e.g. *"Présent"* example in Table A.1; this may also reflect priors as discussed above). We suspect that this may be because training examples have gradients aggregated across an entire passage. Repeating statements that are tangentially relevant can add up to a substantial similarity score, and if better examples receive lower scores due to saturation or other effects, these distractors may rank as top proponents.

## 7.1 DEBUGGING INCORRECT PREDICTIONS

Above, we examined proponent retrievals only for ground truth targets, even when the actual model prediction would have been incorrect. In a model-debugging scenario, however, it may be useful to attribute the *incorrect* predictions of the model so as to understand where this behavior was learned (Grosse et al., 2023; Nguyen et al., 2024) and uncover mislabeled examples or other issues with the training corpus. In Figure 3 (right) we consider the subset of 1592 queries from our evaluation set that the 8B model gets wrong, we retrieve proponents from C4 using TrackStar for the (incorrect) model prediction, and we categorize the proponents using the same scheme as §7.

We observe that a substantial fraction (28.5%) of these answers actually do have entailing passages (7.1% of all proponents in Figure 3 right), and on closer inspection, we find that many of these correspond to alternative correct answers. For example, for *"Ricky Gervais works as: comedian"* the model predicts *"actor"* (he is both), and for *"Victoria Park is located in the following country: Australia"* the model predicts *"United Kingdom"* (there is a Victoria Park in both). However, a much larger fraction of proponents consist of partial matches, suggesting that when the model is wrong, it is often making a guess based on partial information, e.g. using one of the reasoning strategies or priors described in §7.

## 8 CONCLUSION

In this paper, we explore data attribution (*influence*) methods at the scale of C4 pretraining for an 8B-parameter LLM, pushing the frontier of TDA capabilities closer to full pretraining attribution for modern LLMs. Our best method, TrackStar, outperforms previous gradient-based methods both at retrieving examples that entail a fact (*attribution*) as well as examples that influence model predictions (*influence*). Despite this, we find that classical, model-agnostic retrieval methods such as BM25 still perform better on attribution metrics. While this may be partially due to the highly lexical nature of the fact tracing task (Wang et al., 2021; Akyurek et al., 2022), it also demonstrates that attribution and influence may not always be fully aligned, both due to headroom in the method and the fact that different types of non-entailing examples can be influential on a model's prediction. We do find, however, that influence and attribution are more closely aligned as models improve. This suggests that TDA results may even align further with human intuition for newer generations of LLMs, potentially enabling practical usage of this technique to debug model predictions and better understand the connection between training data and model behavior.

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

## A APPENDIX

### A.1 QUALITATIVE EXAMPLES

Examples of top passage retrievals from TrackStar are shown in Tables A.1 and A.2 below.

## Example proponents from TrackStar:

**Présent is in the following language: → English** **(incorrect, groundtruth: French)**
**Proponent retrieval #3** (non-entailing):
Sorry, this entry is only available in Deutsch, English and All Languages. Sorry, this entry is only available in Nederlands, Slovenina, Franais, Polski, English and All Languages. Sorry, this entry is only available in English and All Languages. Sorry, this entry is only available in English and All Languages. Sorry, this entry is only available in English and All Languages. Sorry, this entry is only available in Polski, English and ...

**Victoria Park is located in the following country: → United Kingdom**
**(incorrect\*, groundtruth: Australia)**
**Proponent retrieval #1** (entailing):
Victoria Park in the northern part of Londons East end is 86 hectares of meadows, trees and formal gardens set around two lakes. The Regents Canal runs along the south and west sides of the park and is pleasant to walk along especially on a summer day. The park was donated to the people by Queen Victoria, it was the first public park and opened to the public in 1845. There are a number of good bars and restaurants on the northern edge of the park on Grove Road.

**Ricky Gervais works as: → actor** **(incorrect\*, groundtruth: comedian)**
**Proponent retrieval #1** (entailing):
Going to see Ricky in Toronto tomorrow night, a re-blog felt appropriate. For those not familiar with him, Ricky Gervais is a brilliant British actor, director and writer, best known for portraying David Brent in his original BBC series The Office. A past host of the Golden Globes, the often controversial comedian is known for raising the ire of Hollywood A-listers with his blunt zingers and one-liners. I first noticed him as the museum director Dr. McPhee in A Night at the Museum with Ben Stiller, and later stumbled upon him in The Office on Netflix. I quickly became a fan and adored all his subsequent projects, including; Derek, An Idiot Abroad and Lifes too Short, all hilarious and poignant. As I write this, Gervais new movie Special Correspondents is currently playing on Netflix, and his feature film David Brent Life on the Road will be released in theaters this coming August. Im jealous! I love Ricky. I discovered him in The Office, and I like all his TV shows, while Im still catching up with his movies. Besides his animal rights activism, I love his outspokenness about atheism and in the film The Invention of Lying, he cleverly made fun of the origins of religion. Have a good time tomorrow! Thank you, looking forward to it!

**City is located in the following country: → United States**
**(incorrect\*, groundtruth: Finland)**
**Proponent retrieval #1** (entailing):
Many city is located in Louisiana State, Sabine County and has a unique zip code assigned by the US Postal Service Office. Many Area Code is 318 and Time Zone is Central (GMT -06:00). The US ZIP Code basic format consists of five decimal numerical digits assigned to the Many City. An extended ZIP+4 code, introduced in the 1980s, includes the five digits of the Many ZIP code, a hyphen, and four more digits that determine a more specific location within a given ZIP code in Many, Louisiana. The Many Zip Code is 71449. If you want to send a mail to Many City, you should ...

**Munich Symphony Orchestra originated in the following country: → Germany**
**(correct, groundtruth: Germany)**
**Proponent retrieval #1** (non-entailing):
Amsterdam Baroque Orchestra, conductor Ton Koopman. Label: Challenge Classics. Recording Date: September, 2009.

**Cadillac was formed in the city of: → Detroit** **(correct, groundtruth: Detroit)**
**Proponent retrieval #1** (non-entailing):
RBG: Real Bar Gaming Nerd Jabber Loves... Comics! What did Happy Gilmore do next? 90s nostalgia and ridiculous theories with host Claire Lim, guest host Paul McCallum and special guest Josh Macuga. The podcast is available on iTunes, Spotify or via the embed below.

**Alemayehu Shumye died in the city of: → Addis Ababa**
**(correct, groundtruth: Addis Ababa)**
**Proponent retrieval #1** (non-entailing):
Dr. Alemayehu G. Mariam (Al Mariam) is a professor of political science at California State University, San Bernardino (CSUSB). He received his Ph.D. from the University of Minnesota in 1984, and his J.D. from the University of Maryland in 1988. He serves on the board of the Center for the Study of Hate and Extremism at CSUSB. He has given human rights lectures at the Carr Center, Harvard University, and various law schools including the University of Minnesota, American University and UCLA. He also played a central advocacy role in the passage of H.R. 2003 (Ethiopia Democracy and Human Rights Act) in the House of Representatives in 2007. For the last several years, he has written a weekly web commentary on Ethiopian human rights and African issues. Currently, he has a weekly blog at the Huffington Post and his articles are featured on Pambazuka News, New American Media, Ethiopian Review and Ethio-Media. He has published two volumes on American constitutional law, including American Constitutional Law: Structures and Process (1994) and American Constitutional Law: Civil Liberties and Civil Rights (1998). He is also a Senior Editor of the International Journal of Ethiopian Studies, a leading scholarly journal on Ethiopia.

**Alemayehu Shumye died in the city of: → Addis Ababa**
**(correct, groundtruth: Addis Ababa)**
**Proponent retrieval #2** (non-entailing):
1 Response to "Addis Ababa Bete Alemayehu Eshete" I love this song. Thanks for posting it!!

**Carel Steven Adama van Scheltema had the following native language: → Dutch**
**(correct, groundtruth: Dutch)**
**Proponent retrieval #1** (non-entailing):
Carel Steven Adama van Scheltema was born 26 February 1877 in Amsterdam, North Holland, Netherlands to Frederik Adama van Scheltema (1846-1899) and Hendrika Lulofs (1850-1927) and died 6 May 1924 in Bergen, North Holland, Netherlands of unspecified causes. He married Anna Catharina Kleefstra (1884-1977) 24 October 1907 . Ancestors are from the Netherlands.

Table A.1: Example passage retrievals from TrackStar, from the 8B model over the full C4 corpus. Some model predictions are marked as incorrect\* because they do not match the "ground truth" from the T-REx dataset, but they are still plausibly correct due to either an ambiguous prompt or an alternative correct answer (for example, Ricky Gervais is both an actor and a comedian).

## Top proponent from TrackStar for randomly-sampled queries:

### Thompson River is located in the following country: → Canada
**Proponent retrieval #1**:

Blue Moon. Original Painting by Thompson River artist Carl Stromquist. Carl is deliberate and focused: Deliberate in his desire to paint with excellence, in his quest to earn visions that will generate artwork which honors his Gift of Life and the richness, depth and teachings of the Canadian First Nations. Being Self taught, Carl has spent much time studying the works of renowned Native artisans as well as the timeless truths embodied in the elders' stories and legends. From this journey for truth has come his desire to portray the balance and harmony that exists in the Circle of Life. Carl's art is influenced by his profound love and respect for nature.

### Rachael Ray is a citizen of the following country: → United States
**Proponent retrieval #1**:

The Rachael Ray Show - Snack of the Day!!!! The Rachael Ray Show - Snack of the Day!!!! Copyright 2014 KK's Gourmet. All Rights Reserved.

### Lars Johansson speaks the following language: → Swedish
**Proponent retrieval #1**:

Lars Johan Larsson was born on February 15, 1849. He married Sofia Johansdotter. She was born on March 15, 1847 in Fogels.

### Jean-Baptiste Forest was born in the city of: → Paris
**Proponent retrieval #1**:

Creative Jean-Baptiste Le Divelec had fun turning the entire movie by Stanley Kubrick, 2001: A Space Odyssey, into animated GIFs. It gives 569 GIFs he published on the platform Giphy, through which we can discover the movie second by second, for free. Although the images are mute, he took care of integrating the subtitles in each GIF.

### Disneyland is named after: → Walt Disney
**Proponent retrieval #1**:

Gerlaw City is named after Robert Gerlaw. Gerlaw City is named after Robert Gerlaw, who was born in 1817 and died in 1911. In the early 1850s, Gerlaw came to Warren County and married Marry Black. They moved to Sec. 34 Spring Grove, which became a township in 1854.

### Michel Sardou works as: → actor
**Proponent retrieval #1**:

In late 2012, Michel Sardou began a four-month "Les Grands Moments" tour across France, revisiting his biggest hits in a truly energetic show. Lighting designer Jacques Rouveyrollis used just one type of spotlight on stage throughout the tour - the MAC Viper Profile a somewhat bold choice given it was the lighting fixtures French dbut, but one that delivered on all its promises. A total of 93 MAC Viper Profiles, supplied by the Dushow Group (who currently have 160 in total), were installed, 56 equally spaced over three gantries and 37 on the floor spread across four levels. No traditional spotlights were used a rare occurrence for such a show. The simple lighting plan acted as the sole stage design element and provided the performance area with structure by offering a wide variety of effects. At front of house, 6 MAC III Performances were installed behind the control desk for extra punch. The MAC Vipers enabled Jacques Rouveyrollis to design several different lighting scenarios to accompany the various songs. Using a single type of light source ...

### Ryan Miller is a citizen of the following country: → United States
**Proponent retrieval #1**:

Posted on Tuesday, January 22nd, 2019 at 8:50 pm. Ryan Miller won the Vezina last year and therefor is the top ranked goalie. He also lost in the gold medal game on a questionable shot and got bounced in the first round by the Bruins who then went on to perform one of the most epic collapses in sports history. I not saying he shouldn be number one, I just saying those are things to think about come draft day.. About a year ago, I https://www.nfljerseyscheapcollection.com/ escaped from the overheated frenzy of the Summer Solstice Parade and drove up the canyon, where a breeze whispered through the sycamores and eucalyptuses lining Toro Canyon Creek down toward the ocean below. Amid the serene grace of Myerss Modernist outpost and its surrounding Mediterranean landscape with fruit trees and oaks, I forgot all about the parade. Nobody was visible, and there was nothing to knock on besides girders.. Some of the first moves learned by martial arts students is how to punch and block with the arms. Injuries often occur when a kick is not blocked properly, including bruises, hairline fractures, and even broken bones. To reduce the risk of these types of injuries, a student will wear sparring gear on her arms for her partner and to protect herself as well.. He keeps his office in wholesale jerseys his house, and almost never leaves home, even to pursue the detective work that allows for his expensive lifestyle. Instead, ...

### My Bloody Valentine 3D originated in the following country: → United States
**Proponent retrieval #1**:

No reviews for My Bloody Valentine Complete Edition [Limited Edition] yet. there is a new movie directed by Patrick Lussier. the price for My Bloody Valentine Complete Edition [Limited Edition] drops. there are new images or links available for My Bloody Valentine Complete Edition [Limited Edition].

### Happily Divorced had cast members named: → Fran Drescher
**Proponent retrieval #1**:

Happily Divorced star Fran Drescher is now happily married! The 56-year-old actress said "I do" to Dr. Shiva Ayyadurai, 50, at their beach home in front of a small group of friends and family, People confirms. Drescher shared the news on Twitter on Sunday with a sweet selfie. Ayyadurai, who holds the patent for inventing email, met Drescher a year ago at an event hosted by Deepak Chopra. "Fran heard my talk and we fell in love, and we've been together since that talk," he told the Huffington Post. "Every day is a celebration with Fran. Every day is almost a romantic hangout with her. We're always laughing, always enjoying ourselves." This is the second marriage for The Nanny actress. She was married to Peter Marc Jacobson for 21 years, but divorced him in 1999. Jacobson, who is also Drescher's producing partner, later came out as gay.

### Stadium Beyond the Stars is of the following genre: → science fiction
**Proponent retrieval #1**:

Beyond The Stars: Blade Runner Overview (4 of 4): Do Androids Dream of Electric Sheep? Beyond the Stars is a series of Science Fiction related posts where I discuss different aspects of the genre and the many tropes and plot lines associated with it. Today, I talk about Blade Runners source material, Do Androids Dream of Electric Sheep? in a series of posts focused on the Blade Runner Universe. Beyond Continue reading Beyond The Stars: Blade Runner Overview (4 of 4): Do Androids Dream of Electric Sheep?

Table A.2: Top passage retrieval from TrackStar for randomly-sampled queries with ground truth targets, from the 8B model over the full C4 corpus.

## A.2 METHOD DETAILS

### A.2.1 OUTPUT FUNCTION ABLATIONS

The first step in TrackStar (§3) is to compute the example loss gradient $\nabla_\theta \text{Loss}(z, \theta)$. However, the approximation of the effect of training example $z_m$ on query $z_q$ loss can also be approximated using a different output function $f$ (Park et al., 2023). Specifically, TRAK estimates the effect of $z_m$ on $z_q$ in terms of output function $f$, then converts this effect to an effect on loss by multiplying the projected and corrected gradient vectors by $Q = \frac{\partial \text{Loss}}{\partial f}$. Ideally, $f$ is chosen to be maximally linear with respect to model parameters.

In the case of TRAK specifically, the multi-class margin function $f = \log(\frac{p}{1-p})$ corresponds to $Q = p - 1$. However, TRAK applies $Q$ at the example level, multiplying influence scores by $1 - \bar{p}$, where $\bar{p}$ is the mean probability over tokens in an example (using $1 - p$ rather than $p - 1$ so as not to flip the sign of dot products). TRAK only applies $Q$ to the candidate examples $z_m$; when applying $Q$ at the example level, there is no effect on rankings if $Q$ is applied to the query example $z_q$. Crucially, we find that applying $Q$ at the example level rather than token level significantly hurts performance. MRR scores on the closed set T-REx evaluation (as in §5) drop from 0.122 to 0.001, and recall scores drop from 0.194 to 0.001.

Assuming $Q$ is applied at the token level, we can then evaluate different output functions $f$ in TrackStar. To do this, we (1) take gradients for $f$ in place of loss in Equation 2, and (2) multiply each $G_\theta(z)$ in Equation 2 by $Q = \frac{\partial \text{Loss}}{\partial f}$. We consider three possible output functions:

- Loss $f = -\log(p)$: This is the output function we use in the final version of TrackStar. In this case, no $Q$ term is required.
- Margin (multi-class margin function) $f = \log(\frac{p}{1-p})$: This is the output function recommended by Park et al. (2023) and Engstrom et al. (2024), based on its extension from logistic regression. In this case, $Q = p - 1$.
- Logit $f = \text{Logits}_w$: the logit for each target token $w$. This tests the hypothesis that the target logit is a more linear (with respect to model parameters) and less noisy proxy for loss. In this case, $Q = p - 1$.

We test these output functions with optimizer second moment correction, a non-task-specific Hessian approximation, and unit normalization, i.e. analogous to Experiment 5 in Table 1. Results are reported in Table A.3. We find that varying the output function has fairly little effect, but loss performs slightly better than other output functions.

| Optim. | $f$ | $R$ | Unit norm | MRR | Recall@10 | Tail-patch |
|:---:|:---:|:---:|:---:|:---:|:---:|:---:|
| ✓ | Loss | ✓ | ✓ | **0.295** | **0.406** | **+0.87%** |
| ✓ | Margin | ✓ | ✓ | 0.293 | 0.403 | **+0.87%** |
| ✓ | Logit | ✓ | ✓ | 0.288 | 0.388 | +0.71% |

Table A.3: Results as in Table 1 but evaluating different output functions $f$ to take gradients (Park et al., 2023).

### A.2.2 RANDOM PROJECTION

To efficiently project gradients into lower dimensionality (§3), we use the two-sided random projection from Pruthi et al. (2020) and equivalent to the low-rank projections in Choe et al. (2024). For a gradient matrix $W \in \mathbb{R}^{m \times n}$ projected into dimensionality $d$, rather than use a naive projection of the flattened gradients $P_d \in \mathbb{R}^{d \times mn}$, we use two projection matrices $P_{d_0} \in \mathbb{R}^{\sqrt{d} \times m}$ and $P_{d_1} \in \mathbb{R}^{\sqrt{d} \times n}$. Projection matrix entries are sampled i.i.d. from $\mathcal{N}(0, \frac{1}{\sqrt{d}})$. The resulting projection is:

$$P_{d_0} W P_{d_1}^T \in \mathbb{R}^{\sqrt{d} \times \sqrt{d}} \tag{4}$$

For our models, we concatenate gradients into eight layer blocks. For example, the 8B-parameter model has 32 layers, so we concatenate gradients for every four consecutive layers. We concatenate attention and MLP matrices separately. We then project each concatenated matrix into $d = 4096 =$

$64 \times 64$ dimensions using Equation 4. This results in a total of 2 (attention vs. MLP) $\times$ 8 (layer blocks) $\times$ 4096 dimensions, or $2^{16}$ total dimensions. We ablate over dimensionality in §5.2 by decreasing the dimensionality per layer block.

### A.2.3 TASK-SPECIFIC HESSIAN APPROXIMATION

Our Hessian approximation in §3 follows the Gauss-Newton approximation discussed by Sagun et al. (2018) and Park et al. (2023), which is based on the autocorrelation matrix of the projected gradient vectors. This is computed as $R = \tilde{\Phi}^T \tilde{\Phi}$, where rows of $\tilde{\Phi} \in \mathbb{R}^{n \times d}$ are projected gradient vectors for individual examples. For efficiency and following previous work approximating a block diagonal Hessian (Grosse et al., 2023), we compute the autocorrelation per layer block (recall from §A.2.2 that gradient vectors are projected into 4096 dimensions per layer block). In other words, rather than $\mathbb{R}^{2^{16} \times 2^{16}}$, our Hessian approximations are in $\mathbb{R}^{2^4 \times 2^{12} \times 2^{12}}$. Instead of applying $R^{-1}$ directly in the inner product, we compute $R^{-\frac{1}{2}}$ (this is easily computed from the SVD of $R$) and apply it separately to both the train and query vectors (Eq. 2); this allows us to express retrieval as a symmetric dot product as in Equation 1.

For the task-specific Hessian approximation (§3), we aim to downweight gradient components that are common (often high magnitude) for a given target task. For example, many tasks include a natural language template shared across all examples. To downweight these gradient components, we compute a Hessian approximation $R_{\text{eval}}$ computed from the target task query vectors. However, applying $R_{\text{eval}}$ entirely in place of $R_{\text{train}}$ results in excessive upweighting of components that are rare (often low magnitude) for the target task. If there are relatively few target task examples, $R_{\text{eval}}$ may not even be invertible. Thus, we dampen $R_{\text{eval}}$ as in Equation 3, repeated here for convenience:

$$R = \lambda R_{\text{eval}} + (1 - \lambda) R_{\text{train}}$$

We select $\lambda$ such that roughly the top 1000 out of $2^{16} = 65536$ task-specific components are downweighted. Concretely, we select $\lambda$ such that the $1000^{\text{th}}$ largest singular values of $\lambda R_{\text{eval}}$ and $(1-\lambda) R_{\text{train}}$ are roughly equal. Put another way, we scale $\lambda$ such that the (monotonic) singular value curves of $\lambda R_{\text{eval}}$ and $(1-\lambda) R_{\text{train}}$ intersect at the $1000^{\text{th}}$ component. We note that this requires larger $\lambda$ when $R_{\text{train}}$ is computed over C4 rather than T-REx, because C4 examples have overall larger gradients (and thus larger $R_{\text{train}}$ singular values) due to longer sequence lengths. We set $\lambda = 0.90$ for the T-REx experiments in §5 and $\lambda = 0.99$ for the C4 experiments in §6. We also find that these values work well empirically, selecting from $\{0.50, 0.90, 0.99, 0.999\}$. Determining the optimal $\lambda$ for a given task (i.e. the number of task-specific components to downweight) is an interesting direction for future investigation.

We also note that this task-specific approach defines semantic "dimensions" that should be downweighted (in this case, downweighting examples that support the overall task structure rather than an individual fact). When there is no a priori known task, the non-task-specific Hessian can be used, at a slight cost to performance (Experiment 5 in Table 1). More generally, the determination of semantic dimensions that should be downweighted is somewhat subjective; the eval data itself is not necessarily required for Hessian approximation, but rather any set of examples to define what semantics are less relevant.

### A.3 MODEL DETAILS

As described in §4.1, we pretrain a 154M-, 1B-, and 8B-parameter decoder-only language model on two epochs of English C4 (Raffel et al., 2020). All of our models are implemented using T5X (Roberts et al., 2022). For all model sizes, we use the same SentencePiece tokenizer (Kudo & Richardson, 2018) trained on C4 data with vocabulary size 32K. All model sizes are trained on the same shuffle of the pretraining data. We pretrain with batch size 1024 and sequence length 2048 for two epochs (187K steps). The 154M, 1B, and 8B models reach eval losses (log-perplexities) of 2.34, 1.99, and 1.77 respectively. Specific hyperparameters are in Table A.4.

### A.4 DATASET DETAILS

Our T-REx dataset is merged from KILT (2.3M facts, a further processed version of T-REx; Petroni et al., 2021) and the original T-REx dataset (11.1M facts; Elsahar et al., 2018). These

| Hyperparameter | 154M | 1B | 8B |
|---|---|---|---|
| Layers | 8 | 16 | 32 |
| Embedding size | 1024 | 2048 | 4096 |
| Hidden size | 1024 | 2048 | 4096 |
| MLP hidden size | 4096 | 8192 | 16384 |
| Attention heads | 4 | 8 | 16 |
| Attention head size | 256 | 256 | 256 |
| Optimizer | | | Adafactor |
| Learning rate | | | 0.01 |
| Vocabulary size | | | 32K |
| Batch size | | | 1024 |
| Sequence length | | | 2048 |
| Activation function | | | SwiGLU |
| Attention type | | | Multi-query |
| Position embedding | | | RoPE |
| Learning rate decay | | | Inverse square root |
| Warmup steps | | | 10K |
| Dropout | | | 0.0 |

Table A.4: Language model pretraining hyperparameters, following Chowdhery et al. (2023).

datasets consist of fact triples (input entity, relation, target entity) scraped from Wikipedia. We start from the KILT dataset because it has useful surface form aliases (different possible surface strings) for each entity, for more robust scoring of open-ended text generation. We exclude entity aliases with less than three characters. However, the KILT dataset does not directly contain entity URIs and entailing sentences from Wikipedia abstracts. Thus we match the KILT dataset back with the original T-REx dataset using entity string matching, keeping only facts that are unambiguously matched between the two datasets. The original T-REx dataset contains entity URIs and machine-annotated entailing Wikipedia abstracts. We remove ambiguous facts that have multiple correct target URIs (e.g. "*France*", "*shares border with*", "*Spain, Germany, etc*"), along with nine manually-identified ambiguous or noisy relation types: `facet_of`, `is_a_list_of`, `instance_of`, `located_in_the_administrative_territorial_entity`, `part_of`, `subclass_of`, `has_part`, `main_subject`, `residence`. The resulting dataset has 1.2M fact triples covering 97 relation types.

To obtain entailing Wikipedia sentences from the annotated abstracts for each fact, we use the sentence boundaries, entity spans, and relation spans annotated in T-REx. Each abstract in T-REx contains annotated fact triples, with corresponding input entity, target entity, and relation spans. Thus, within an entailing abstract for a fact, we mark an individual sentence as entailing if the input, target, and relation spans all fall within the sentence boundaries. We do not include entity spans that match a stop word (e.g. "*she*"), because these cases are generally ambiguous pronoun coreferences to a previous sentence, and thus the sentence in isolation does not entail the fact. However, because abstracts in T-REx often have un-annotated entity spans, we also mark a sentence as entailing within an entailing abstract if at least one surface form of each entity appears within the sentence boundaries, based on lowercased string matching; while this slightly biases our entailment annotations towards lexical matching, we observe that the sentence annotations have a large number of false negatives if we omit this step. We use our annotations of entailing sentences for fact tracing evaluations in §5.

For C4 frequency counting, we use lowercased string matching, marking a C4 example as relevant to a fact if it matches at least one surface form alias for both entities in the fact (Kandpal et al., 2023). Frequencies range from zero to roughly $10^6$ out of 365M examples in C4.

Finally, because standard existing prompt templates are designed for masked rather than autoregressive language models (Petroni et al., 2019), we manually write a natural language template for each relation type. Templates all end with a colon, which we find better constrains the model to answer the fact rather than continue the sentence in another way during open-ended generation. For exam-

ple, the template for `country` is "*[entity0] is located in the following country:*". Results for other templates are reported in §A.5.1.

For all reported experiments, we use the same subsample of 5.4K facts balanced for fact frequency. Specifically, we separate facts into six frequency buckets: 1 to 10, 10 to $10^2$, $10^2$ to $10^3$, $10^3$ to $10^4$, $10^4$ to $10^5$, and $10^5$ to $10^6$ occurrences in C4, with frequency annotations described above. We randomly sample up to 1K facts from each frequency bucket. Per bucket, we restrict facts with a given relation and target entity (e.g. "*country*", "*USA*") to 25 examples, and we restrict each target and relation overall to 100 examples. We also restrict samples that are incorrect for all three model sizes to 100 per bucket. The first five frequency buckets successfully sample 1K facts each. The highest frequency bucket only samples 415 facts, because there are overall fewer of those high-frequency facts.

## A.5 ADDITIONAL RESULTS

### A.5.1 RESULTS FOR DIFFERENT PROMPT TEMPLATES

To verify that our results are not entirely reliant on the templates used for factual prompts, we write two additional prompts for each factual relation type. As described in §A.4, our original templates are designed to constrain model generations to produce a factual completion, e.g. "*[entity0] was born in the city of:*". Our second set of templates removes the colon and words designed only to constrain the generation, e.g. the template "*[entity0] was born in*". The last set of templates rewords the prompts such that the input entity is not at the beginning of the prompt, e.g. the template "*The birthplace of [entity0] is*".

Results for different templates are reported in Table A.5. In line with the results in the main text, for all templates, BM25 and Gecko perform better than TrackStar for attribution (MRR and recall), but TrackStar performs better for influence (tail-patch scores over $2\times$ higher).

| Template | Method | MRR | Recall@10 | Tail-patch |
|----------|--------|-----|-----------|------------|
| Original | T-REx gold | Gold | Gold | +0.52% |
|          | BM25 | 0.592 | 0.773 | +0.41% |
|          | Gecko | **0.620** | **0.794** | +0.31% |
|          | TrackStar | 0.365 | 0.496 | **+0.90**% |
| Variation #1 | T-REx gold | Gold | Gold | +0.55% |
|          | BM25 | **0.617** | **0.797** | +0.57% |
|          | Gecko | 0.593 | 0.766 | +0.30% |
|          | TrackStar | 0.331 | 0.460 | **+1.16**% |
| Variation #2 | T-REx gold | Gold | Gold | +0.39% |
|          | BM25 | **0.603** | **0.791** | +0.22% |
|          | Gecko | 0.579 | 0.760 | +0.01% |
|          | TrackStar | 0.299 | 0.424 | **+0.79**% |

Table A.5: Results as in Table 1 but using different templates for factual prompts.

### A.5.2 TAIL-PATCH RESULTS FOR TOP-*k* PROPONENTS

In Table 1, we report the tail-patch score (target probability increase from a train step on a single proponent) averaged over the top $k = 10$ proponents for each fact. In Table A.6, we report tail-patch scores when averaging over the top $k = 1, 3, 5$, and 10 proponents per fact. As expected, lower $k$ leads to higher tail-patch scores, because only higher-ranked proponents are considered when $k$ is lower. For all $k$, TrackStar outperforms BM25, Gecko, and the "ground truth" entailing sentences from T-REx, in line with the results in the main text.

### A.5.3 RESULTS SPLIT BY MODEL CORRECTNESS

In §5.1, our results are based on proponents retrieved for the ground truth target for each factual prompt. However, the model only correctly predicts a subset of these facts. Intuitively, we might expect results to differ based on whether proponents are retrieved for facts that the model predicts

| Method | $k=1$ | $k=3$ | $k=5$ | $k=10$ |
|--------|-------|-------|-------|--------|
| T-REx gold | +0.89% | +0.71% | +0.61% | +0.52% |
| BM25 | +0.81% | +0.59% | +0.50% | +0.41% |
| Gecko | +0.94% | +0.59% | +0.45% | +0.31% |
| TrackStar | **+1.90%** | **+1.40%** | **+1.18%** | **+0.90%** |

Table A.6: Tail-patch scores as in Table 1 but averaged over the top $k$ proponents for different $k$. Results in the main text use $k=10$.

correctly vs. incorrectly, even though the proponents are retrieved for the ground truth correct answer in both cases. We split these two conditions in Table A.7.

We find that whether the model predicts a fact correctly does not substantially affect MRR and recall, although incorrectly-predicted facts tend to have slightly higher scores. This may be because correctly-predicted facts are more likely to have diminished or saturated gradients, because the model already assigns high probability to the correct output (Pruthi et al., 2020). Indeed, incorrectly-predicted facts are much easier to tail-patch to higher probabilities (tail-patch scores; Table A.7) than facts that the model already predicts correctly. The model-agnostic methods (BM25 and Gecko) also exhibit this effect, suggesting that our proponents are not necessarily better for incorrectly-predicted facts; for those facts, it is simply easier to push the model probability towards the correct prediction because the model does not already have a high correct probability. In both cases, trends between methods are consistent.

| Exp. # | Optim. | $R$ | Unit | MRR | Recall@10 | Tail-patch |
|--------|--------|-----|------|-----|-----------|------------|
| T-REx gold | – | – | – | Gold | Gold | **+0.32%** / **+1.10%** |
| BM25 | – | – | – | 0.591 / 0.593 | 0.780 / 0.756 | +0.30% / +0.66% |
| Gecko | – | – | ✓ | **0.611 / 0.641** | **0.789 / 0.806** | +0.21% / +0.54% |
| TRAK | | ✓* | | 0.001 / 0.001 | 0.001 / 0.001 | −0.02% / −0.02% |
| 1 | | | | 0.055 / 0.086 | 0.099 / 0.149 | +0.24% / +0.61% |
| 2 | | | ✓ | 0.258 / 0.286 | 0.350 / 0.376 | +0.48% / +1.07% |
| 3 | | ✓ | ✓ | 0.282 / 0.312 | 0.387 / 0.428 | +0.64% / +1.36% |
| 4 | ✓ | | ✓ | 0.291 / 0.321 | 0.405 / 0.432 | +0.53% / +1.16% |
| 5 | ✓ | ✓ | ✓ | 0.286 / 0.316 | 0.397 / 0.428 | +0.65% / +1.41% |
| TrackStar | ✓ | Mixed | ✓ | **0.358 / 0.379** | **0.489 / 0.515** | **+0.69%** / **+1.42%** |

Table A.7: Results from Table 1 split into facts that the model predicts correctly vs. incorrectly (left / right). Trends between methods are similar, but tail-patch scores are higher for facts that the model predicts incorrectly. Intuitively, it is easier to push the model probability towards the correct prediction when the model is not already correct.

## A.6 ADDITIONAL TASKS

To facilitate further TDA research, we provide retrievals from TrackStar on the 8B-parameter LLM for several additional tasks, including factual predictions, factual errors, commonsense reasoning, arithmetic, and open-ended generation. Specifically:

- **T-REx ground truth** (Elsahar et al., 2018; Petroni et al., 2021): This is the task used for the main results in this paper. We use the T-REx dataset of fact triples and manually write templates, as described in §A.4. The 8B model accuracy for this task is 32.4% using open-ended generation (chance close to 0.0%). TDA queries use the factual prompt as input text and the ground truth answer as target text. As described in §A.4, we use a sample of 5415 factual queries.

- **T-REx incorrect predictions**: This uses the same set of prompts as the T-REx ground truth queries. We filter to facts that the 8B model predicts incorrectly (1593 facts), and we manually remove facts that are clearly ambiguous (e.g. as in Table A.1) or where the model prediction does not follow the task template (e.g. just producing or repeating a sentence rather than responding to the factual prompt). This results in a set of 966 incorrectly-predicted facts. TDA queries use the factual prompt as input text and the incorrect model prediction as target text.

- **COPA** (Roemmele et al., 2011): We use the 400 commonsense reasoning completions in the COPA train set. The 8B model accuracy for this task is 80.3% (assigning higher probability to the correct completion; chance 50.0%). TDA queries use the input prompt as input text and the ground truth commonsense completion as target text.

- **PIQA** (Bisk et al., 2020): We use the 1838 physical commonsense reasoning completions in the PIQA validation set. The 8B model accuracy for this task is 78.3% (assigning higher probability to the correct completion; chance 50.0%). TDA queries use the input prompt as input text and the ground truth commonsense completion as target text.

- **Arithmetic word problems** (Roy & Roth, 2018): We use the 1492 arithmetic word problems from Roy & Roth (2018). We note that because our models are only trained on C4 (a smaller and less curated dataset than most modern LLMs), the 8B model performs quite poorly on the arithmetic word problems (1.6% accuracy using open-ended generation). However, its top proponents for ground truth answers generally consist of examples with mathematical operations and similar word problems. TDA queries use the input question with "*Answer:*" appended as the input text and the ground truth answer as target text.

- **Simple arithmetic**: We use templates to generate simple arithmetic prompts (e.g. "5+5="). We generate 500 queries each for integer addition, subtraction, multiplication, and division. We sample the first operand from [1, 100] and the second operand from [1, 10]. As with the arithmetic word problems, the 8B model performs quite poorly on simple arithmetic prompts (9.7% accuracy using open-ended generation), but its top proponents for ground truth answers generally consist of examples containing arithmetic. TDA queries use the input prompt as input text and the ground truth answer as target text.

- **Story generation**: We use templates to generate story prompts for 50 manually-compiled story genres (e.g. "*fantasy*", "*scifi*", and "*horror*"). The templates are of the form "*Genre: [genre]. [Book/Film] summary:*". We generate four story summaries per genre: two book summaries (sampling temperatures 0.3 and 0.7) and two film summaries (sampling temperatures 0.3 and 0.7). We find that the resulting 200 story summaries are fairly coherent, usually consisting of a short paragraph describing a generic story of that genre. TDA queries use the input prompt as input text and the generated story as target text.

For each query, we retrieve the top proponents from C4 using TrackStar. For the added tasks (i.e. the non-T-REx tasks), we use the non-task-specific Hessian approximation $R$ (Experiment 5 in Table 1) because the number of dimensions to downweight is both somewhat subjective and task-dependent (§A.2.3). Our data and the results browser to view our identified influential pretraining examples are at https://github.com/pair-code/pretraining-tda.

