# OpenReview forum: "Scalable Influence and Fact Tracing for Large Language Model Pretraining"
_ICLR.cc/2025/Conference — ICLR 2025 Poster_

### Official Review · Reviewer_D1Kg · 2024-10-25

**Soundness:** 4
**Presentation:** 4
**Contribution:** 3
**Rating:** 8
**Confidence:** 3

**Summary:**

The paper proposes TrackStar, a scalable, gradient-based influence tracing method for large language model (LLM) pretraining. It aims to identify influential training data examples for model predictions, enhancing model transparency. To achieve TrackStar, the authors compute the influence between two examples as the dot product between the projected and
corrected model gradients for those examples. TrackStar outperforms prior influence methods in identifying examples that entail a fact (attribution) and influence predictions (causal impact), though classical retrieval methods like BM25 still excel at attribution. They further analyze the effectiveness of TrackStar using extensive experiments.

**Strengths:**

- The paper addresses a critical gap in training data attribution (TDA) by scaling to pretraining setups, which may be further used for efficient pre-training or transparent training.
- Experiments are well-designed, covering a range of models (up to 8B parameters) and extensive datasets, including T-REx and C4, to validate TrackStar’s effectiveness.
- The authors offer a detailed analysis of the distinction between influence and attribution, adding clarity to how models use training data.

**Weaknesses:**

- ''For each fact, there are an average of 2.5 “ground-truth” sentences out of 19.6M sentences total''. Does this mean any training or testing examples contain 2.5 relations? The statement needs further clarity.
- In the experiments, although the pass rate of records extracted by StarTrack is lower than that of BM25, training a model on StarTrack's examples leads to greater improvement. I wonder if limiting the selection to only the top 3 or top 5 examples—or even just the top example—would reverse this outcome.
- For C4 evaluation, an NLI model is used. However, the accuracy cannot be fully guaranteed since C4 consists of longer documents. Human annotation of a subset is needed.

**Questions:**

Please see the weaknesses. I have no more questions. I will improve my rate if you can address my concerns

---

> ### Author Response · Authors · 2024-11-21
>
> Thank you for the great suggestions!
>
> > "For each fact, there are an average of 2.5 'ground-truth' sentences out of 19.6M sentences total". Does this mean any training or testing examples contain 2.5 relations? The statement needs further clarity.
>
> We have updated the wording to: "For each fact, there are an average of 2.5 'ground-truth' *entailing* sentences out of 19.6M sentences total." Each fact is a single relation, but on average 2.5 of the 19.6M T-REx sentences entail a given fact. For example, the fact ("The Old Man and the Sea", "author", "Ernest Hemingway") has relation type "author"; an entailing T-REx sentence is "The Old Man and the Sea is a short novel written by the American author Ernest Hemingway in 1951 in Bimini, Bahamas, and published in 1952".
>
> > In the experiments, although the pass rate of records extracted by StarTrack is lower than that of BM25, training a model on StarTrack's examples leads to greater improvement. I wonder if limiting the selection to only the top 3 or top 5 examples—or even just the top example—would reverse this outcome.
>
> Great question! Tail-patch results for k=1, 3, and 5 are now reported in Appendix A.4.2. In all cases, TrackStar has a tail-patch score over 2x higher than BM25, in line with the original result.
>
> > For C4 evaluation, an NLI model is used. However, the accuracy cannot be fully guaranteed since C4 consists of longer documents. Human annotation of a subset is needed.
>
> The entailment model we use is trained for factual consistency evaluation, including training on summarization data with longer premises (https://aclanthology.org/2023.emnlp-main.127/), and 49.0% of C4 passages have length <256 tokens. We have also updated our results to split input C4 passages by sentences (using regex matching, taking the maximum entailment score for a sliding window of three sentences) when evaluating whether a C4 passage entails a fact (Section 4.3). We find that this alleviates the effect where longer C4 passages had high false-positive entailment rates.
>
> Based on blinded manual annotation of 100 C4 passages marked as entailing and non-entailing respectively using this method, we find a false positive rate of 13% and a false negative rate of 11%. Of the incorrect annotations, most (54%) are semantically related passages that give a high likelihood of a fact but may not strictly entail it. For example, a passage describing a famous Austrian person does not imply that they were from Vienna, but it is statistically likely.

---

> > ### Comment · Reviewer_D1Kg · 2024-11-22
> >
> > Thank you for the detailed feedback. The experimental results addressed my concerns, and I will increase my score to 8.

---

### Official Review · Reviewer_wLd8 · 2024-11-03

**Soundness:** 2
**Presentation:** 2
**Contribution:** 3
**Rating:** 6
**Confidence:** 3

**Summary:**

The paper studies extending influence functions to LLMs by modifying key components of TRAK (Park et al., 2023) such gradient computation, correction, and input processing (e.g., vector normalization). The authors applied their approach, dubbed TRAKSTAR, to a decoder-only model with 3 different sizes pretrained on the C4 dataset and compare to TRAK and other retrieval based approach. They found their approach to perform better than influence-based method, but to perform much worse compared to retrieval-based approaches.

This leads them to investigate the mist-alignment between attribution and influence (Section 7, the part that I find most interesting in the paper) and show that LLMs rely on priors for predictions---not just entailment. They perform an analysis on error cases.

**Strengths:**

* I like how the paper discusses difference/discrepancy between attribution and influence. I think this discussion could be useful to future work.
* I found the error analysis in section 7 to be interesting.
* Different design choices are well-ablated.
* The proposed approach outperforms influence-based baselines at retrieving influential examples.

**Weaknesses:**

* The tail-patch metric used to evaluate influence will probably depend on the templates used. It is unclear how reliable such metric is.
* If I understood correctly, the hessian approximation proposed requires access to an evaluation dataset. The authors should explain how this would work in the case of no have access to the full evaluation dataset.
* It seems to me that lots of "hacks" are needed to get the approach to work (e.g., gradient correction, normalization, gradient compression via projection etc.) But I suppose this is a limitation of influence functions rather than of the proposed technique.
* It is unclear to me that scaling influence functions to LLM is practical/reliable at all, as it seems that spurious correlations between the input fact and the pretraining examples to play a big role. For instance, looking at the errors provided in the Appendix, many of the retrieved proponents seem to completely unrelated to the fact-- some don't even have any word overlap such as example ID 862.

**Questions:**

* You mention that you manually write natural language templates for all 97 relation types. I wonder how much will the results rely on these templates? Do you have experiments with different template variations?
* Could you provide a discussion on real-world scenarios where scaling influence functions to LLM would be helpful/practical?

---

> ### Author Response · Authors · 2024-11-21
>
> Thank you for the great questions!
>
> > The tail-patch metric used to evaluate influence will probably depend on the templates used. It is unclear how reliable such metric is… I wonder how much will the results rely on these templates? Do you have experiments with different template variations?
>
> This is a good point! We have added results with two additional template variations to Appendix A.4.1. For example, the "place of birth" factual relation now has three templates: "[entity0] was born in the city of: [entity1]", "[entity0] was born in [entity1]", and "The birthplace of [entity0] is [entity1]". For all templates, our main trends hold: classical retrieval methods (BM25 and Gecko) perform substantially better for attribution (MRR and recall@10), but TrackStar performs over 2x better for influence (tail-patch score).
>
> > If I understood correctly, the hessian approximation proposed requires access to an evaluation dataset. The authors should explain how this would work in the case of no have access to the full evaluation dataset.
>
> Yes, the task-specific Hessian approximation requires access to an evaluation dataset. This task-specific approach defines semantic "dimensions" that should be downweighted (in this case, downweighting examples that support the overall task structure rather than an individual fact). When there is no a priori known task, the non-task-specific Hessian can be used, at a slight cost to performance (Experiment 5 in Table 1). More generally, the determination of semantic dimensions that should be downweighted is somewhat subjective – it's not so much that we need the eval data for approximation as that we use a set of examples to define what semantics are most relevant. As a topic for future work, it might be possible to use synthetic data to induce this as well.
>
> > It seems to me that lots of "hacks" are needed to get the approach to work (e.g., gradient correction, normalization, gradient compression via projection etc.) But I suppose this is a limitation of influence functions rather than of the proposed technique.
>
> Each of the proposed methods is grounded in previous work, practical motivations, and theoretical justification:
> * Gradient second moment and Hessian correction are theoretically motivated to align gradient dot products with change in loss (https://arxiv.org/abs/2303.14186). This also approximates the Fisher information metric between model gradients.
> * Gradient compression via projection is necessary to reduce dimensionality enough to store an index of gradients for efficient retrieval. Gaussian random projection has theoretical guarantees from the Johnson-Lindenstrauss lemma, and it has been applied in previous work (https://arxiv.org/abs/2002.08484, https://arxiv.org/abs/2303.14186, https://arxiv.org/abs/2402.04333, see also dimensionality ablations in Section 5.2).
> * Unit normalization is motivated as $\ell$-relative influence that constrains overall loss change (https://arxiv.org/pdf/2003.11630), reducing the contribution of "loud" examples with high gradient norm.
>
> > It is unclear to me that scaling influence functions to LLM is practical/reliable at all, as it seems that spurious correlations between the input fact and the pretraining examples to play a big role. For instance, looking at the errors provided in the Appendix, many of the retrieved proponents seem to completely unrelated to the fact-- some don't even have any word overlap such as example ID 862.
>
> The examples in Table A.1 were selected to demonstrate types of proponents we observed – in particular, to show headroom/error patterns – and they are not necessarily representative of the distribution of proponents. We have added Table A.2, where we report the top proponent for a random sample of facts. Across all 5K facts, the top proponent has word overlap with the input or target entity for 84.4% of facts, and there is at least one such proponent in the top 10 for 94.9% of facts.
>
> > Could you provide a discussion on real-world scenarios where scaling influence functions to LLM would be helpful/practical?
>
> Scaling TDA / influence functions to LLMs can be useful for data curation (selecting pretraining data to target particular evaluation tasks, e.g. https://arxiv.org/pdf/2401.12926, https://arxiv.org/abs/2402.04333), interpreting underlying causes of model predictions (e.g. https://arxiv.org/pdf/2405.13954, https://arxiv.org/abs/2308.03296), and diagnosing misbehaviors (https://arxiv.org/abs/2409.16978) by identifying mislabeled or harmful training examples. As a concrete example, when debugging incorrect factual predictions in our work, we uncovered several ambiguous facts in T-REx. For example, the prediction that Victoria Park is in the United Kingdom (Table A.1) is labeled as incorrect by T-REx, but the top proponents from TrackStar show that there actually is a Victoria Park in the United Kingdom described in several pretraining examples.

---

> > ### Comment · Reviewer_wLd8 · 2024-11-25
> >
> > Thanks for your responses and additional results. I will maintain my currently positive score.

---

### Official Review · Reviewer_wust · 2024-11-04

**Soundness:** 3
**Presentation:** 3
**Contribution:** 3
**Rating:** 6
**Confidence:** 3

**Summary:**

This paper introduces a method for training data attribution that works effectively at scale. The proposed method relies on gradient-based estimations, where the authors combine several techniques such as optimizer state correction and normalized encodings to improve its attribution performance. Experiments on an 8B model pretrained on a corpus of over 160 billion tokens show the effectiveness of the proposed method and its different design choices. There are also some other interesting findings such as the misalignment between examples that can fully entail the target and examples that have actual high influence on model prediction.

**Strengths:**

- Training data attribution is very important for understanding the connection between training data and the model's learned behaviors, and scalability is one key criterion especially nowadays when the model/data are large. The method this paper introduces is shown to be both more scalable and more effective than previous methods.

- The ablation experiments are thorough and clearly show the effectiveness of the various design components of the proposed method.

- The findings/analysis on the misalignment between factual attribution and causal influence are very interesting and could be a good inspiration for the community to understand LLM's behaviors around factual prediction.

**Weaknesses:**

- The different design components of the proposed method are mostly adaptations of previously proposed ones, whereas the proposed method is a combination of them. Hence, the technical novelty of this work is a bit limited.

- Overall, the proposed designs are quite heuristical and not very solidly justified. For example, why *can* we enforce a block-diagonal structure for the auto-correlation matrix or perform unit normalization on the input vectors? What are the assumptions for doing these and are there some theoretical basis for them?

**Questions:**

See weaknesses.

---

> ### Author Response · Authors · 2024-11-21
>
> Thank you for the helpful comments!
>
> > The different design components of the proposed method are mostly adaptations of previously proposed ones, whereas the proposed method is a combination of them. Hence, the technical novelty of this work is a bit limited.
>
> Our work combines previous approaches under one framework, and our evaluations demonstrate which components are necessary to scale gradient-based influence methods to realistic settings for LLM pretraining. As far as we are aware, our work is the first to run quantitative evaluations of influence methods at LLM pretraining scale (>1B-parameter models, unfiltered pretraining corpora).
>
> > Overall, the proposed designs are quite heuristical and not very solidly justified. For example, why can we enforce a block-diagonal structure for the auto-correlation matrix or perform unit normalization on the input vectors? What are the assumptions for doing these and are there some theoretical basis for them?
>
> The block-diagonal structure in the autocorrelation matrix is enforced for computational tractability, in line with previous works (e.g. https://arxiv.org/abs/2308.03296), although in future work it may be possible to relax this. The unit normalization is based both on empirical results (https://arxiv.org/abs/2205.12600, https://arxiv.org/abs/2402.04333, https://arxiv.org/abs/2205.11482, https://arxiv.org/abs/2405.13954) and theoretical justification as $\ell$-relative influence (details in https://arxiv.org/pdf/2003.11630). Theoretically, $\ell$-relative influence identifies training examples that maximize the query example loss change while constraining the overall loss change.

---

> > ### Comment · Reviewer_wust · 2024-11-26
> >
> > Thank you for the response, which addresses my concerns about justifications for the design choices. I still hold the first weakness point regarding technical novelties. Overall, I would like to raise the score but I feel a score of 8 is not adequate, and hence I would maintain my current score.

---

### Official Review · Reviewer_kPtE · 2024-11-05

**Soundness:** 3
**Presentation:** 4
**Contribution:** 3
**Rating:** 8
**Confidence:** 3

**Summary:**

The authors propose TrackStar, a gradient-based influence method which works effectively on a 8B LM pretrained on 160B tokens, without the need for filtering or subsampling. The proposed method performs well at identifying instances which influence the model prediction, but model agnostic methods suc as BM25 still outperform it at finding passages which contain explicitly relevant facts. The paper delves into the problem of misalignment between factual attribution and causal influence, which they show becomes a smaller issue when scaling models.

**Strengths:**

- Well written and motivated, easy to follow, very comprehensive exprimental evaluation with ablations and subsequent error analyses
- Scales up TDA to large models, corpora and number of targets
- Identifies and demonstrates a misalignment between attribution and influence
- Empirically shows that influence becomes closer to attribution when scaling models

**Weaknesses:**

The only drawback I see is not delving deeper into the mismatch between infuelnce and attribution, however this is understandable as it is worthy of a separate paper.

**Questions:**

Tail-patch probability increase works worse with model scale (Fig 2). Do you believe this to be caused by a single instance mattering less for a larger model, or due to the inherent better performance of the underlying model? It would be interesting to distribute these scores for cases where the underlying model was right/wrong in its prediction.

---

> ### Author Response · Authors · 2024-11-21
>
> Thank you for the insightful comments!
>
> > Tail-patch probability increase works worse with model scale (Fig 2). Do you believe this to be caused by a single instance mattering less for a larger model, or due to the inherent better performance of the underlying model? It would be interesting to distribute these scores for cases where the underlying model was right/wrong in its prediction.
>
> We include results split by whether the model was right or wrong in Appendix A.4.3. Indeed, we find that tail-patch scores are over 2x higher for incorrectly-predicted facts vs. correctly-predicted facts. For incorrectly-predicted facts, it appears easier to push the model probability towards the correct prediction because the model does not already have a high correct probability. Thus, the effect where larger models have lower tail-patch scores is likely due at least partially to the higher performance of the larger models (higher "correct" probabilities before tail-patching). We also note that the tail-patch scores for the 8b model do not appear to have plateaued relative to projection dimensionality (Figure 2), which also likely contributes to its lower tail-patch scores. To control for these potential confounds, we only compare tail-patch scores within a given model size.

---

> > ### Comment · Reviewer_kPtE · 2024-11-27
> >
> > Thank you for the response. I will maintain my score as it reflects my feelings regarding the paper well.

---

### Author Response · Authors · 2024-11-21

Thank you all for the thoughtful reviews, and for the recognition of rigorous quantitative evaluations, ablations, and implications for TDA in LLMs!

Reviewers wust and wLd8 asked about the theoretical motivation of the different components of our approach, and how these relate to previous work. A major contribution of our work is showing that these address different aspects of the problem:
* Gradient second moment and Hessian correction are theoretically motivated to align gradient dot products with change in loss (https://arxiv.org/abs/2303.14186). This also approximates the Fisher information metric between model gradients.
* Gradient compression via projection is necessary to reduce dimensionality enough to store an index of gradients for efficient retrieval. Gaussian random projection has theoretical guarantees from the Johnson-Lindenstrauss lemma, and it has been applied in previous work (https://arxiv.org/abs/2002.08484, https://arxiv.org/abs/2303.14186, https://arxiv.org/abs/2402.04333, see also dimensionality ablations in Section 5.2).
* Unit normalization is motivated as $\ell$-relative influence that constrains overall loss change (https://arxiv.org/pdf/2003.11630), reducing the contribution of "loud" examples with high gradient norm.

Reviewers wLd8 and D1Kg also suggested that our results might be dependent on design choices, specifically prompt templates and k values for top-k tail-patch analyses. We have added results for additional prompt templates and k values to the Appendix (also described below). For all variations, our main results hold.

---

### Meta-Review · Area_Chair_RTUv · 2024-12-20

**Metareview:**

Summary:

This paper studies training data attribution (TDA), which aims to attribute model outputs back to specific training examples, and it has been challenging to apply existing methods to the full scale of LLM pretraining. This paper introduces a gradient-based method that allows retrieving influential examples for an 8B-parameter language model from a pretraining corpus of over 160B tokens with no need for subsampling or pre-filtering. The proposed method combines several techniques, including optimizer state correction, a task-specific Hessian approximation, and normalized encodings. The paper also identifies a misalignment between factual attribution and causal influence, and conducts an in-depth analysis around it. It finds that influence more closely aligns with attribution, with increasing model size and training tokens. The paper also examines different types of examples identified as influential by the proposed method, and has interesting findings that while many directly entail a particular fact, others support the same output by reinforcing priors on relation types, common entities, and names.

Strengths:

1. Reviewers agree that the problem being studied in this paper (training data attribution) is very important, and the method this paper proposes is shown to be more scalable and more effective than previous methods overall.

2. Reviewers applaud the paper’s efforts conducting a very comprehensive evaluation with ablations and error analyses.

3. Reviewers also point out that the findings and analysis around the misalignment between factual attribution and causal influence are very interesting.

Weaknesses:

Some standing issues after the rebuttal period:

Reviewer wust still has some concern about the technical novelty (i.e., different design components of the proposed method are mostly adaptations of previously proposed ones, and the proposed method in this paper is a combination of them). But this seems to be a relatively minor issue, as this paper does make significant contributions by putting everything in a framework and conducting quantitative evaluations and ablations at the LLM pretraining scale.

**Additional Comments On Reviewer Discussion:**

Weaknesses in the original reviews that were (partially) addressed during the rebuttal period:

1. Reviewer wust’s concern on the design choices being heuristic and not very well justified. The authors has addressed this concern with further explanations and by drawing connections with recent work.

2. Most clarification questions were addressed during the discussion period.

---

### Decision · Program_Chairs · 2025-01-22

Accept (Poster)